# Trajectories of Biological Values and Vital Parameters: An Observational Cohort Study of Adult Patients with Sickle Cell Disease Hospitalized for a Non-Complicated Vaso-Occlusive Crisis

**DOI:** 10.3390/jcm8091502

**Published:** 2019-09-19

**Authors:** Raphael Veil, Simon Bussy, Vincent Looten, Jean-Benoît Arlet, Jacques Pouchot, Anne-Sophie Jannot, Brigitte Ranque

**Affiliations:** 1Medical Information Department, Georges Pompidou European Hospital, AP-HP, 75015 Paris, France; vincent.looten@gmail.com (V.L.); annesophie.jannot@aphp.fr (A.-S.J.); 2INSERM U1138, University Paris Descartes, Sorbonne University, 75006 Paris, France; 3LPSM, CNRS UMR 8001, Sorbonne University, 75005 Paris, France; simon.bussy@gmail.com; 4Internal Medicine Department, Sickle Cell Referral Center, Georges Pompidou European Hospital, AP-HP, 75015 Paris, France; jean-benoit.arlet@aphp.fr (J.B.-A.); jacques.pouchot@aphp.fr (J.P.)

**Keywords:** humans, adults, red cells, hemoglobinopathies, anemia, sickle cell, bio-monitoring, hospitalization

## Abstract

Hospital admission of patients with sickle-cell disease (SCD) presenting with a vaso-occlusive crisis (VOC) can be justified by pain refractory to usual outpatient care and/or the occurrence of a complication. Yet, the trajectories of vital parameters and standard biomarkers throughout a non-complicated VOC has not been established. In this observational cohort study, we describe the course of routine parameters throughout 329 hospital stays for non-complicated VOC. We used a new spline-based approach to study and visualize non-specific time-dependent variables extracted from the hospital clinical data warehouse. We identified distinct trends during the VOC for hemoglobin level, leukocytes count, C-Reactive Protein (CRP) level and temperature. Hemoglobin decreased after admission and rarely returned to steady state levels before discharge. White blood cell counts were elevated at admission before immediately decreasing, whereas eosinophils increased slowly throughout the first five days of the stay. In over 95% of non-complicated VOC-related stays, the CRP value was below 100 mg/L within the first day following admission and above normal after 48 hours, and the temperature was below 38 °C throughout the entire stay. Knowing the typical trajectories of these routine parameters during non-complicated VOC may urge the clinicians to be more vigilant in case of deviation from these patterns.

## 1. Introduction

Sickle-cell disease (SCD) is the most common genetic hematological disorder worldwide [1], resulting from monogenic autosomal recessive transmission. Over 2% of the world population carries the variant [2] and 1.12 births out of 1000 are affected with the homozygous form of the disease [3], making SCD an increasing burden for the upcoming decades [4]. Although multiple genotypes can lead to the phenotypical trait, homozygous S-hemoglobin (SS) is accountable for roughly 70% of cases [5]. The inherited mutated variant leads to a defective β-hemoglobin sub-unit, which polymerizes under certain conditions, thus sickling the patient’s erythrocytes [6,7,8]. The sickled red blood cells can obstruct capillaries, predisposing patients to acute ischemia to downstream organs and tissues [7,8]. Such episodes, called vaso-occlusive crisis (VOC), are responsible for recurrent acute pain syndromes lasting for a few hours to a few days. SCD also results in chronic hemolysis, leading to chronic anemia and vasculopathy and ultimately to organ damage, as well as an increased susceptibility to infection partly due to functional asplenia [9,10,11,12,13,14].

Most VOCs are managed by adult patients at home using acetaminophen and mild oral opioids [15,16]. Yet, since VOCs can vary in terms of clinical severity, home management can be extremely difficult at times for some patients. Hospital admission for VOC is justified by refractory pain to available ambulatory analgesics and/or by the occurrence of a complication such as an infection or an acute chest syndrome. The diagnostic of VOC is retained in SCD patients for any pain lasting more than 2-4 hours and having no other explanation than SCD. The pain is usually localized to the limbs or spine. While this diagnosis is easily made and accurate in most cases, it remains impossible to differentiate VOCs from rare cases of pain of other origins, such as neuropathic pain, acute exacerbations of chronic pain from avascular necrosis, psychological pain, or opioid-use disorder, which is a possible complication of recurrent pain killer use. Although it has been observed that VOCs are often associated with increased hemolysis (elevated lactate dehydrogenase (LDH) and bilirubin), worsening anemia (low hemoglobin level) and moderate systemic inflammation (elevated C-Reactive Protein (CRP) and hyperleukocytosis), there is currently no reliable diagnostic biomarker of an on-going VOC. The abovementioned common inflammatory biomarkers are monitored to detect the occurrence of an infectious complication during a VOC, but the usual range of values for vital parameters and common laboratory results at the time of diagnosis and during the course of non-complicated VOCs are unknown. In hospitalized VOCs, intravenous opioid-based patient-controlled analgesia (PCA) is a first choice treatment to control pain for adult patients; therefore, the resolution of the VOC is corroborated by the decrease of both pain intensity and opioid requested doses. In such context, establishing the common range of vital parameters and usual laboratory results at admission and during the course of a non-complicated VOC episode could help the clinician either rule out the diagnosis of VOC or detect a possible complication. As a result, it could be a first step towards guidelines for monitoring VOC episodes, to determine which laboratory tests to perform and when.

Today, clinical data warehouses (CDW) can automatically store heterogeneous real-life data from electronic health records (EHR), allowing researchers to use clinical, administrative, and biological retrospective data to answer these kinds of questions. The Georges Pompidou University Hospital (GPUH) in Paris, France, has set up such a data warehouse [17]. The GPUH internal medicine department is also a national referral center for SCD adult patients and manages over 300 VOC-related hospitalizations per year. The reuse of health data from this department, facilitated by the CDW, could help describe how biomarkers and vital signs behave throughout a hospitalized non-complicated VOC.

The main objective of this study is to describe the behavior of biomarkers and vital parameters and to highlight their trends throughout a non-complicated VOC hospital stay.

## 2. Materials and Methods

This is a single center retrospective cohort study. Data were extracted from GPUH CDW, which uses the i2b2 star-shaped standard [18,19]. It contains routine care data from the GPUH EHR, such as clinical records, administrative data, laboratory results and medical imaging reports. The CDW is divided into several categories, including demographics, vital signs, diagnoses from the International Classification Disease codes (ICD-10), procedures, clinical data from structured questionnaires, free text reports, biological test results, and drug prescriptions.

The sample included all hospital stays from patients admitted to the internal medicine department for VOC (labeled with ICD-10 code D57.0) between 1 January 2010 and 31 December 2015. We excluded patients coded as suffering from opioid use disorder (ICD-10 F11) as well as those who were treated with either Methadone or Buprenorphine (which in France are exclusively used for the management of opioid use disorder), both of which were confirmed by hospitalization reports and drug prescriptions. We also excluded complicated VOCs, defined as stays for which:

The patient stayed in the Intensive Care Unit (ICU) at any point during the hospitalization;

The stay’s severity was rated as 3 or 4 on the 4-level French *CMA* (‘*Complications ou Morbidité Associée*’) scale of severity (see Appendix A for details);

The patient received at least one red blood cell transfusion;

The stay was associated with a diagnosis of complication (e.g., bacterial infection, thrombosis, etc.; see Appendix B for details; in our sample, acute chest syndromes, which were identified based on textual reports since they are absent of the ICD-10, were excluded de facto by previous criteria; VOCs with an isolated fever were not considered to be a complicated VOC and were not excluded from our cohort);

The duration of the stay was higher than 90th percentile of duration of the remaining stays, after applying all of the excluding criteria listed above; since it is uncommon for non-complicated VOC stays to be prolonged, we added this criterion in order to exclude stays with a potentially interfering clinical condition (complication, comorbidity, etc.).

To ensure complete reporting of our routinely collected health data, we followed the reporting of studies conducted using observational routinely-collected health data (RECORD) statement [20]. This study received approval from the Institutional review board of Georges Pompidou University Hospital (IRB 00001072-project n°CDW_2014_0008) and the French data protection authority (CNIL-n°1922081).

Structured data (such as demographic data, admission and discharge dates and timestamps, laboratory tests, vital parameters and opioid prescriptions) where automatically pulled from the EHR, whereas unstructured data found in free text reports had to be manually extracted. An exhaustive list of all extracted and derived variables is available in Appendix C.

Patients’ characteristics were summarized using mean and interquartile range for quantitative variables and frequencies for binary variables. We also grouped patients by genotype and tested differences between the groups: using the Chi-squared test (or Fisher’s exact test) for categorical variables, and Wilcoxon’s sum-rank test for quantitative variables.

Regarding selected biomarkers and vital parameters, each variable’s mean trajectory and its confidence interval were estimated using a procedure described in Appendix D. This procedure allows for the automation of averaged trajectory visualization for any type of repeatedly measured variable, even if the number of measures and the time between them varies from one stay to another.

The Python code used to generate the figures is available online in the form of annotated programs, together with notebook tutorials and all generated figures [21].

## 3. Results

### 3.1. Population Description

Three hundred and twenty-nine hospital stays for non-complicated VOC were included (Figure 1) for a total of 164 patients. The number of stays per patient varies from 1 to 10 (Figure A1a in Appendix E), with a median of 2 stays per patient. The duration of the stay ranged from 1 to 10 days, with an average of 4.4 days (Figure A1b in Appendix E). Patients’ statistics at the first included stay showed some differences between the SS genotype group vs. other genotypes (Table 1).

### 3.2. Trends in Laboratory Results

#### 3.2.1. Complete Blood Count

Hemoglobin values ranged from 4 to 14 g/dL at hospital admission and slowly decreased from 9.5 g/dL on average down to 8.5 g/dL after the fifth day following admission (Figure 2). By comparison, the median [IQR] hemoglobin level at steady state was 9 [8,10] g/dL. The hematocrit- and reticulocytes-related trends were similar to those of hemoglobin (Figure A2 and Figure A3 in Appendix E). As expected, females had lower hemoglobin values throughout their stay than males (Figure A4 in Appendix E).

The white blood cell count showed an early spike over 12.10^9^/L around admission, and then decreased and stabilized at around 10.10^9^/L after the 2nd day of the stay (Figure 3a). The average neutrophil count displayed a similar trend (Figure 3b), whereas average lymphocyte, monocyte, and basophil counts showed no clear trend during the stay. Average eosinophil count increased from 2.10^8^/L at admission to 4.10^8^/L around the 5th day of the stay (Figure 3c).

The average platelets levels remained roughly stable throughout the stays, between 300.10^9^/L and 350.10^9^/L (Figure A5 in Appendix E).

#### 3.2.2. Other Laboratory Results

Average CRP rapidly increased in the first 2 days after admission, and then stabilized around 60 mg/L before decreasing slowly (Figure 4). Notably, after the 2nd day following hospital admission, fewer than 5% of VOCs showed a normal CRP level (Figure A6a in Appendix E). Conversely, within the first 24 hours after admission, less than 5% of episodes reached the 100 mg/L CRP threshold (Figure A6b in Appendix E).

The average LDH levels mildly decreased during the stays, from 450 U/L at baseline, to 400 U/L after the 5th day of hospital stay (Figure A7 in Appendix E). Accordingly, the average direct bilirubin level accordingly decreased from 40 μmol/L at baseline to less than 20 μmol/L after the 7th day of the stay (Figure A8 in Appendix E).

The average serum protein level decreased from 76 g/L to 72 g/L within the first day (Figure A9 in Appendix E).

No clear trend emerged for serum electrolytes (sodium, chloride, potassium, total calcium and bicarbonates), renal function markers (creatinine, urea, and glomerular filtration rate estimation according to the ‘modification of diet in renal disease’ (MDRD) formula), alkaline phosphatase, and liver function markers (aspartate and alanine transaminases). All other figures relating to laboratory results are available online [21].

### 3.3. Trends in Vital Parameters

Despite showing clear day/night cycles, the patient’s average temperature remained stable throughout the stay at around 37 °C (Figure 5), with a slight difference between males and females (Figure A10a in Appendix E). Additionally, less than 5% of patients ever reached the 38 °C temperature threshold throughout the entire stay (Figure A10b in Appendix E).

No clear trend emerged for blood pressure, heart rate, peripheral oxygen saturation (under oxygen therapy) and respiratory rate. All other figures relating to vital parameters are available online [21].

## 4. Discussion

In this study, we described average trends (and 95% confidence intervals) of routine laboratory results and vital parameters during non-complicated VOC-related hospital stays, using longitudinal data analysis through the use of routine care data. GPUH applies a no-paper policy and our CDW pulls data from a large variety of sources. Moreover, our data was enriched with manually extracted as well as derived variables. Thus, we analyzed a large number of variables, whether they were important from an expert point of view or just routinely monitored. We were able to highlight: a slow and limited decrease of the hemoglobin level (about −1 g/dL during the stay); an initial increase of the white blood cell and neutrophil counts immediately followed by a decrease; and a progressive increase of the eosinophil count. The knowledge of these results may help clinicians assess the diagnosis of VOC, in particular when it comes to distinguishing between VOC and unrelated pain (e.g., neuropathic pain, acute exacerbations of chronic pain from avascular necrosis, somatoform pain due to psychological stress, or pain from opioid-use disorder). These ranges could also be useful to raise the possibility of a complication (e.g., infection or acute chest syndrome) during the course of a VOC if the patient’s vital parameters or laboratory results lay outside of them. Although most vital parameters and biomarkers we monitored during VOCs filled very wide ranges of values, some of them followed distinct trajectories that could help in the assessment of a possible differential diagnosis, or of the likelihood of a complication in case of deviation from these trajectories.

Therefore, several conclusions can be drawn from our results:

Regarding VOC diagnosis, mean hemoglobin level did not differ from values observed at steady state. Most patients displayed a significant inflammatory syndrome with elevated CRP and hyperleukocytosis, but these markers’ trajectories were not distinct enough to draw any conclusion regarding the diagnosis of VOC in case of normality during the first 48 hours of the stay. In contrast, after 48 hours following admission, only 5% of VOC episodes displayed a normal CRP value, and this percentage decreased overtime. Thus, in case of normal CRP value after 48 hours, it seems reasonable to consider the possibility of a differential diagnosis. Notably, the eosinophil count rose significantly, which is in line with the previously described activated state of eosinophils in SCD patients [22]. This could also be a consequence of subclinical adrenal insufficiency, a frequent phenomenon under opioid therapy [23,24].

Regarding VOC complication, the CRP increase observed in the first days following admission is in line with a previous study that described CRP trajectory in such a context [25]. However, within the first 24 hours after admission, a CRP value over the 100 mg/L threshold was observed in less than 5% of non-complicated VOCs. Such a result suggests that when the CRP level at admission is above this threshold, it might be reasonable bring up the possibility of an associated infection or an acute chest syndrome. The same reasoning goes for temperature levels, which remained under 38 °C from admission to discharge for 95% of non-complicated VOC-related stays, suggesting that other diagnoses may be considered as a cause of the fever.

Regarding VOC resolution, the hemolysis markers and hemoglobin level did not get back to steady state levels before hospital discharge; thus, they are not useful in assessing VOCs resolution. It is worth noting that the CRP level starts decreasing after the 2nd day following admission, although it does not return to a normal value before hospital discharge. The absence of anemia resolution back to the steady state hemoglobin level could be a consequence of lowered reticulocyte production due to systemic inflammation, which delays the renewal of red blood cells after the episode of increased hemolysis.

This study benefitted from the tool we developed to analyze longitudinal data stemming from the reuse of routine care data. Such data have become increasingly available thanks to the development of electronic records coupled with CDW, but are difficult to handle as they are measured at different times from one patient to another. This is why former studies either focused on one or two parameters, with longitudinal measures [25,26], or considered many parameters but then settled on measures at admission [27] or at steady state [28]. Instead, our procedure allows for a broader automated graphical description of any time-dependent variable repeatedly measured during the timeframe of interest. When used on routine care data, automatically displaying averaged trajectories of time-dependent variables could help visually detect trends among them. Therefore, it is an interesting data-mining tool for studying large amounts of longitudinal processes. We would also argue that such advantages would only gain in relevance over time, with the quick deployment of EHR and CDW in various hospitals throughout developed countries.

Nevertheless, our study presents several limits: 

Albeit the fact that it was not the objective of our study, we were unable to produce sensitivity/specificity analysis nor comparative statistics, since we could not identify an appropriate control group, due to the heterogeneity of CVO complications (acute chest syndrome, infections, thrombosis, refractory pain, etc.).

Because of a lack of a gold standard criterion, the diagnosis of non-complicated VOCs was inferred from several criteria to exclude both complicated VOCs and non-VOC related pain episodes. By restricting the diagnostic criteria too much, we might have induced a diagnostic bias. This is particularly true regarding isolated fever, which can sometimes be considered as a sign of a complication. We manually verified that among the hospital stays that had been excluded from the study because of an ICD-10 code of sepsis, none were due to an isolated fever without any documented infection.

As it was performed in a single SCD referral center cohort, our study’s results should be considered exploratory until reproducibility is confirmed. The fact that the GPUH internal medicine department is an SCD center of expertise could potentially induce a selection bias compared with usual VOC-related hospitalization and early care practices. Similarly, since biological and vital monitoring was prescribed by experts, there is possibly a measurement bias compared with usual VOC-related monitoring practices.

Due to lack of data regarding the period of self-treatment priori to seeking medical care, we could not assess the impact of this time-lapse on the measured variables. However, in our urban region, thanks to the ease and free access to medical care in hospitals for patients with SCD, the time between the beginning of the CVO pain and the admission to the emergency unit rarely exceeds 12 hours.

Our patients are managed using a very standardized protocol, including intravenous hydration, folic acid, intravenous morphine delivered via a PCA device and systematic acetaminophen and nefopam three times a day (unless there is an intolerance to one of these drugs, which is rare). The systematic use of acetaminophen may have contributed to the quasi absence of fever observed during the hospital stay.

## 5. Conclusions

Our paper describes the mean trajectories of routine biomarkers and vital parameters monitored during hospitalization for non-complicated VOC, using a procedure allowing for the automation of averaged trajectory visualization for any type of repeatedly measured variables. The most interesting variables, namely hemoglobin, leucocytes (and more specifically eosinophils), CRP and temperature, were quickly identified by the presence of distinct trends when looking at averaged trajectories. We also performed above/under threshold proportion analysis on visually selected variables: in over 95% of non-complicated VOC-related stays, the CRP value was below 100 mg/L within the first day following admission and above normal after 48 hours, and the temperature was below 38 °C throughout the entire stay. Knowing the typical trajectories of routine parameters during non-complicated VOC may urge the clinicians to be more vigilant in case of deviation from these patterns, since it could indicate a diagnostic error or a complication. Yet, global assessment through clinical expertise remains essential in the management and surveillance of VOCs.

A similar study performed on a multicenter cohort from EHR and CDW equipped centers, with different levels of expertise, could potentially strengthen and complement our results as well as confirm their reproducibility.

## Figures and Tables

**Figure 1 jcm-08-01502-f001:**
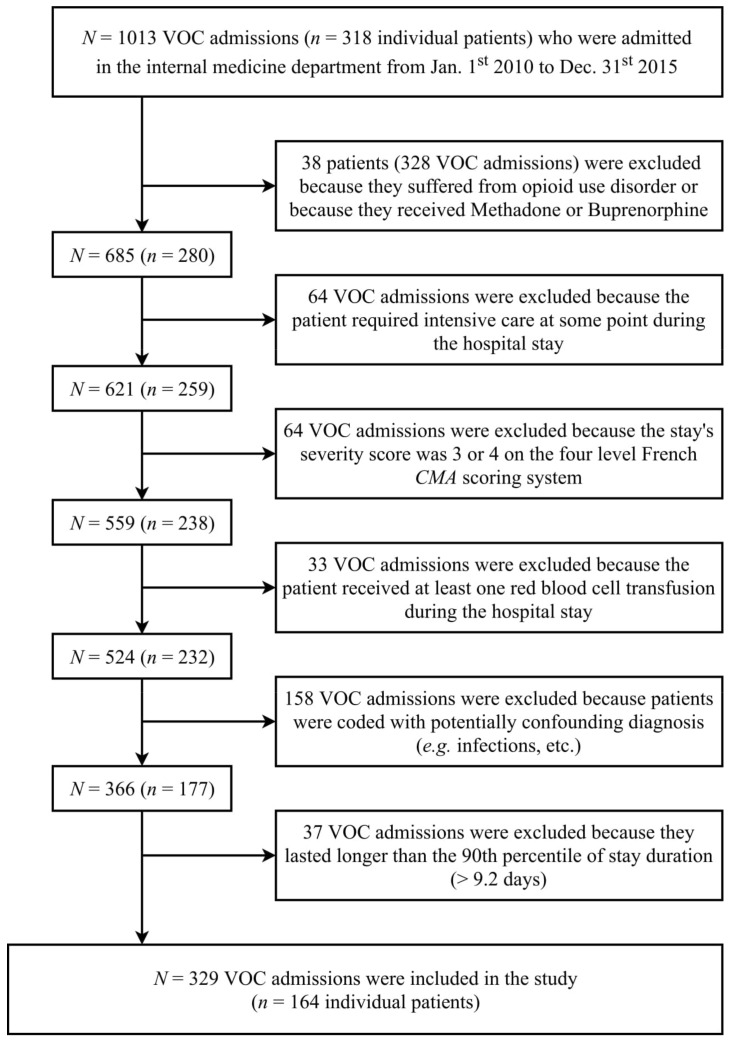
Flowchart.

**Figure 2 jcm-08-01502-f002:**
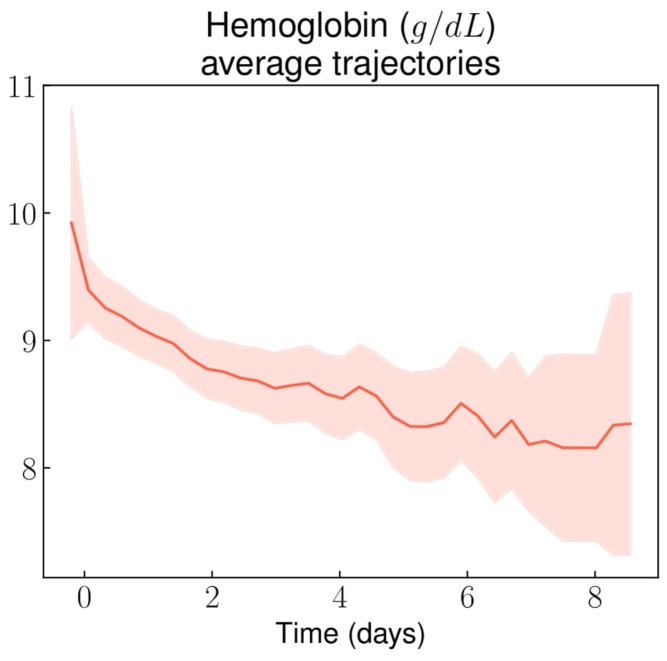
Hemoglobin average trajectory throughout non-complicated VOC-related hospital stays.

**Figure 3 jcm-08-01502-f003:**
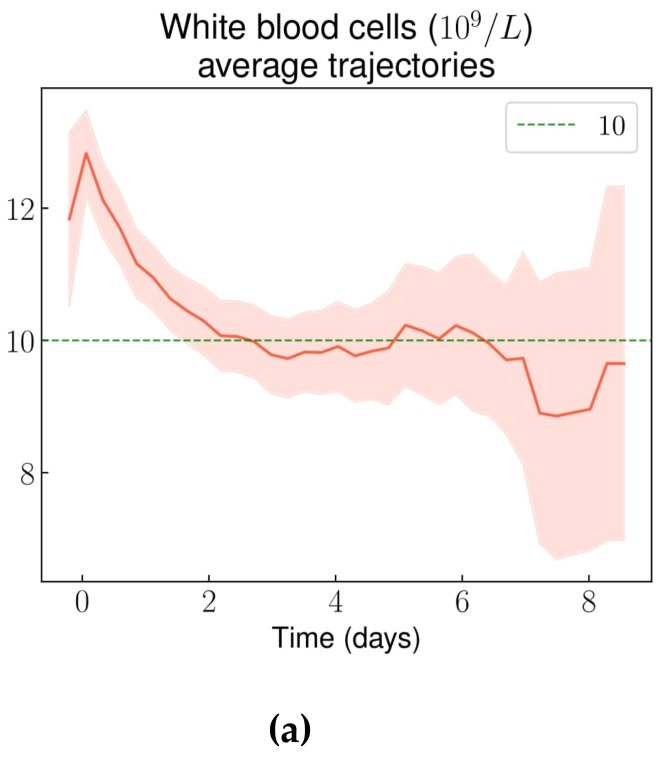
White blood cells: (**a**) white blood cell count average trajectory throughout non-complicated VOC-related hospital stays; (**b**) Neutrophil count average trajectory throughout non-complicated VOC-related hospital stays; (**c**) Eosinophil count average trajectory throughout non-complicated VOC-related hospital stays.

**Figure 4 jcm-08-01502-f004:**
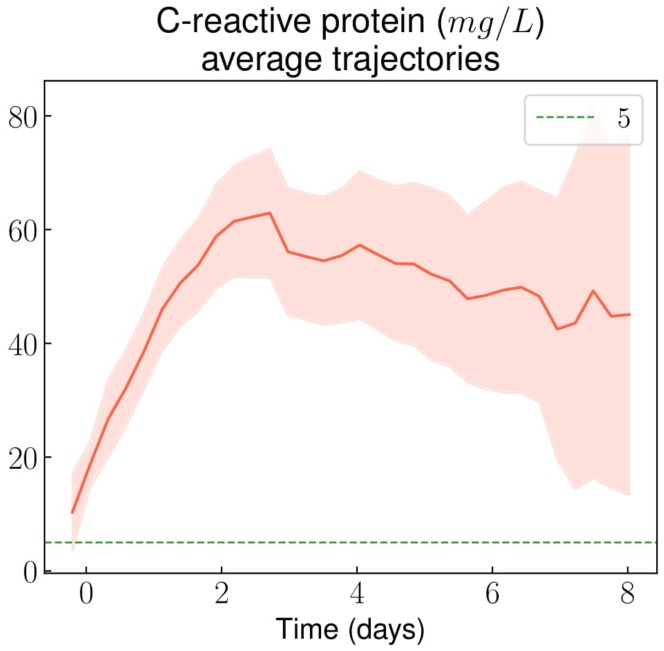
C-reactive protein (CRP) average trajectory throughout non-complicated VOC-related hospital stays.

**Figure 5 jcm-08-01502-f005:**
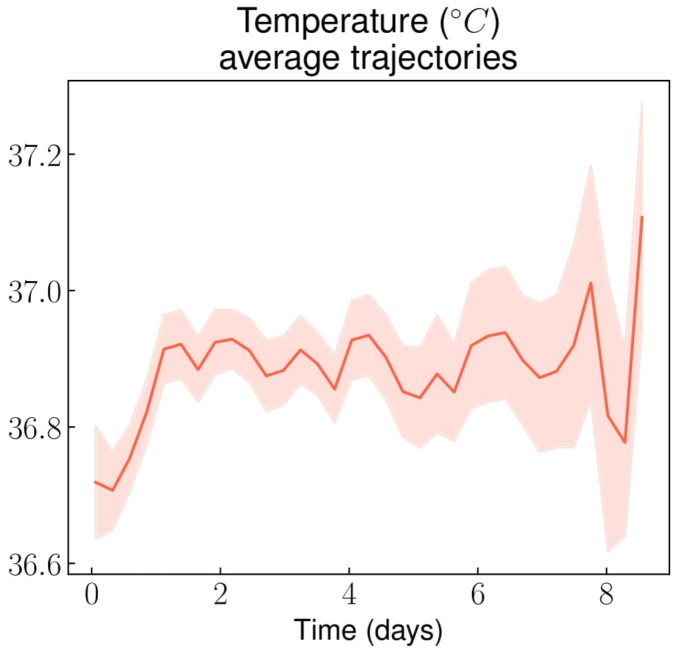
Temperature average trajectory throughout non-complicated VOC-related hospital stays.

**Table 1 jcm-08-01502-t001:** Population description and comparison of patients’ characteristics between genotypes.

	Whole Sample	SS Genotype	Other SCD Genotype ^4^	*p*-Value ^2^
All included patients	*n*^3^ = 164	*n*^3^ = 121 (74.8%)	*n*^3^ = 43 (26.2%)	
Females	87 (53.1%)	63 (52.1%)	24 (55.8%)	0.724
Age at first hospital admission	27 [22; 34]	26 [22; 32]	33 [21; 39]	0.041
Steady state hemoglobin (g/dL)	9 [8; 10]	8 [8; 9]	10 [9.5; 11]	<0.001
Acute chest syndrome ^1^	111 (67.7%)	93 (76.9%)	18 (41.9%)	<0.001
Avascular bone necrosis ^1^	37 (22.6%)	23 (19%)	14 (32.6%)	0.089
Retinopathy ^1^	19 (11.6%)	12 (9.9%)	7 (16.3%)	0.275
Leg skin ulceration ^1^	10 (6.1%)	9 (7.4%)	1 (2.3%)	0.457
Ischemic stroke ^1^	6 (3.7%)	3 (2.5%)	3 (7%)	0.186
Dialysis ^1^	2 (1.2%)	1 (0.8%)	1 (2.3%)	0.457
Pulmonary hypertension ^1^	3 (1.8%)	2 (1.6%)	1 (2.3%)	1
Male patients only	*m*^3^ = 77	*m*^3^ = 58 (75.3%)	*m*^3^ = 19 (24.7%)	
Priapism ^1^	13 (16.9%)	13 (22.4%)	0 (0%)	0.030

^1^ As part of the medical history at the time of the first included stay; ^2^ The *p*-values are obtained from univariate testing for differences between groups. ^3^
*n* is the number of patients and *m* is the number of male patients. Quantitative data are displayed as ‘Median [IQR]’. Qualitative data are displayed as ‘Number of patients (proportion)’. Steady state hemoglobin is the patient’s ‘usual’ hemoglobin level, outside of a vaso-occlusive crisis (VOC) episode, as pulled from all available free text reports. ^4^ ‘Other SCD genotype’ refers to heterozygosity for hemoglobin S and another β-globin chain abnormality, typically hemoglobin C or β-thalassemia+.

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
