# Peer review of "Trajectories of Biological Values and Vital Parameters: An Observational Cohort Study of Adult Patients with Sickle Cell Disease Hospitalized for a Non-Complicated Vaso-Occlusive Crisis"

_jcm, 2019, doi:10.3390/jcm8091502_

Round 1

Reviewer 1 Report

The authors report on the Trajectories of biological values and vital parameters for non complicated VOC s in Adult SCD patients. They perform the study using a clinical data warehouse that they developed in their center. I believe this is a very important aspect of their work since it's a crucial issues in chronic genetic disorders lasting many years and with numerous clinical events (both acute and chronic) in the life of a patient. There is indeed difficulty in funding, storing and analyzing data regarding clinical events and natural history and solutions to answer unmet needs are really needed. I have a few suggestions:

1) The paper refers to VOC in adults: I would clearly state this in the Title ("...study of adult patients with SCD...") and also throughout the text when they describe treatment of VOC (page 22 line 54, page 2 line 70-PCA is NOT the first choice in children or adolescents)

2)Introduction: There are several specifications that could be made: page 2 line 53-54: please cite reference regarding the data that most VOC are managed at home by ADULT patients with acetaminophen and mild oral opioids. While this might be true in their center it is not so in other places. DO not generalise. Line 57-58: the authors should define VOC for the purpouise of this manuscript, considering that VOC are usually defined as "pain lasting more than 2-4 hours and having no other explanation than SCD". therefore I believe that what they state that it is impossible to differentiate pain of VOC from other pains is not true and should be revised

3) In the methods it would be important to better detail the CDW

4) How many SB° patients were there? Why were they NOT included together with SS patients? This is the regular approach in all studies.(See table 1)

5) Why were patients with pain crisis lasting more than 90th %ile excluded?

6) the results are well described and I have no observations

7) Could the authors better describe some practical implications for clinicians of their study?

Reviewer 2 Report

The article "trajectories of biological values and vital parameters: an observational cohort study of patients with sickle cell disease hospitalized for non-complicated vaso-occlusive crisis" aims to trend laboratory values and vitals to serve as biomarkers during a hospitalized non-complicated vaso-occlusive crisis to contribute to future management guidelines.

Overall, the article was thorough, clear, and well organized in its language and objective. Strengths of the article and research include stringent and well-defined inclusion and exclusion criteria and the large data set data was obtained from. There are a few specific and general areas of clarification or adjustment that the reviewer would like to call to attention:

Line 53: For many patients, VOCs are very difficult to manage at home. Perhaps stating that VOCs range in clinical severity and may necessitate inpatient hospitalization.

Line 57: While vaso-occlusive crises often is described as bone pain, it does not "rely" on specifically bone pain as a diagnostic criteria nor does the pain need to be localized to the extremities or spine. 

Line 58: The diagnosis of a VOC is not straightforward to make, as the second part of this sentence states it is often impossible to differentiate the diagnosis from other pain syndromes. 

In the hospitalizations analyzed, there must be variation in the number of data points (ie laboratory values) for each patient. How is this accounted for? While most providers would obtain a complete blood count during a non-complicated VOC admission, there is no standard for obtaining all of the values analyzed (LDH, CRP, Direct Bilirubin). Did all hospitalized patients have these values and the same number of values? If so, did they all have them drawn at the same time from admission? What about patients who did not have all of these values, how were they accounted for in the data analysis? 

It is difficult to draw conclusions off of predicted trajectories and not measured values, particularly when its stated that average length of hospitalization was 4 days, though all graphs go out to 10 days.

Figure 5 shows an uptrending temperature graph at the end of 10 days however it is stated that temperature was table throughout hospitalization.

There is no mention of variations of management for these vaso-occlusive crises.  Perhaps if certain pharmacologic measures were taken (ie acetaminophen or ibuprofen were used) in the management of pain, that may have affected vitals and/or laboratory results as well. 
